# Prognostic Analysis of HPV Status in Sinonasal Squamous Cell Carcinoma

**DOI:** 10.3390/cancers14081874

**Published:** 2022-04-08

**Authors:** Alexandre Tendron, Marion Classe, Odile Casiraghi, Hélène Pere, Caroline Even, Philippe Gorphe, Antoine Moya-Plana

**Affiliations:** 1Head and Neck Oncology Department, Gustave Roussy Cancer Campus, Université Paris Saclay, 94805 Villejuif, France; alexandre.tendron@gustaveroussy.fr (A.T.); caroline.even@gustaveroussy.fr (C.E.); philippe.gorphe@gustaveroussy.fr (P.G.); 2Molecular Radiotherapy and Therapeutic Innovation, UMR 1030, 94805 Villejuif, France; marion.classe@gustaveroussy.fr; 3Pathology Department, Gustave Roussy Cancer Campus, Université Paris Saclay, 94805 Villejuif, France; odile.casiraghi@gustaveroussy.fr; 4Virology Department, European Hospital Georges Pompidou, 75015 Paris, France; helene.pere@aphp.fr; 5Functional Genomics of Solid Tumors (FunGeST), INSERM, Centre de Recherche des Cordeliers, Paris and Sorbonne Université, 75006 Paris, France; 6Inserm U981, Melanoma Group, Gustave Roussy Cancer Campus, Université Paris Saclay, 94805 Villejuif, France

**Keywords:** HPV, sinonasal cancer, squamous cell carcinoma, nasal cavity, overall survival, disease-free survival

## Abstract

**Simple Summary:**

Sinonasal squamous cell carcinoma (SNSCC) is a rare and aggressive malignancy with poor prognosis. Human papilloma virus (HPV) can induce SNSCC although its incidence and impact on outcomes remains unclear. We performed a retrospective cohort study of patients with SNSCC, HPV status being determined with p16 immunohistochemistry followed by RNA in situ hybridization. Fifty-nine patients were included. RNAscope was positive in nine cases (15.2%). Patients with HPV+SNSCC were younger (*p* = 0.0298) with a primary tumor originating mainly in nasal fossa (*p* < 10^−4^). Among patients who were curatively treated, overall survival was better for HPV+SNSCC (*p* = 0.022). No prognostic value of p16 expression was reported. Patients with HPV+SNSCC have better oncologic outcomes, probably due to earlier tumor stage with primary location predominantly in the nasal fossa, a more suitable epicenter to perform a surgical resection with clear margins. P16 expression seems to not be a good surrogate of HPV status in SNSCC.

**Abstract:**

Sinonasal squamous cell carcinoma (SNSCC) is a rare and aggressive malignancy with poor prognosis. Human papilloma virus (HPV) can induce SNSCC although its incidence and impact on patients’ outcomes remains unclear. We performed a retrospective cohort study of patients with SNSCC treated consecutively in a comprehensive cancer center. HPV status was determined with p16 immunohistochemistry followed by RNA in situ hybridization (RNAscope). The incidence, clinical characteristics, and oncologic outcomes of HPV+SNSCC were assessed. P16 prognostic value was evaluated. Fifty-nine patients were included. Eleven (18.6%) SNSCC were p16+ with five (8.4%) doubtful cases. RNAscope was positive in nine cases (15.2%). Patients with HPV+SNSCC were younger (*p* = 0.0298) with a primary tumor originating mainly in nasal fossa (*p* < 10^−4^). Pathologic findings were not different according to HPV status. Among patients who were curatively treated, overall survival was better for HPV+SNSCC (*p* = 0.022). No prognostic value of p16 expression was reported. Patients with HPV+SNSCC have better oncologic outcomes, probably due to earlier tumor stage with primary location predominantly in the nasal fossa, a more suitable epicenter to perform a surgical resection with clear margins. P16 expression seems not to be a good surrogate of HPV status in SNSCC.

## 1. Introduction

Sinonasal malignancies are rare and aggressive entities with a poor prognosis. Indeed, they represent only 3–5% of head and neck cancers, and less than 1% of all malignant tumors [1,2,3]. The most represented histological type is squamous cell carcinoma (SCC) (35–60%) [1,4,5], whose prognosis is poor with an overall survival of 34–40% at five years [6,7,8]. The prognosis has not evolved in recent years despite technical progress both in surgery and radiotherapy [1,6,7,8]. Multimodal strategy combining chemotherapy, surgery, and adjuvant chemoradiotherapy is recommended [9,10]. Surgical resection with clear margins of these frequently locally advanced tumors is difficult to achieve, given the complex anatomy of the region [11]. Thus, local control remains challenging with local recurrences representing the main oncologic event [6,11,12].

Human Papillomavirus (HPV) is a well-known carcinogen for oropharyngeal SCC (OPSCC). However, unlike tobacco-related OPSCC, HPV-induced tumors constitute a sub-group with high chemo-radiosensitivity and more favorable prognosis [13,14]. Today, given this particularly good prognosis, several therapeutic de-escalation protocols are underway to limit the therapeutic morbidity [15,16,17].

Although the presence of HPV has been demonstrated in the sinonasal tract, its implication in SNSCC is still poorly described [13,18]. As HPV+OPSCC is clinically different from HPV-, the same findings may be observed for HPV+SNSCC [19]. Moreover, the potential chemo-radiosensitivity of these tumors could also constitute a prognostic factor in SNSCC. Many studies have searched for the presence of HPV in sinonasal tumors, but detection techniques were heterogenous [18,20,21]. Currently, several HPV detection techniques are performed, indirectly by looking for p16 or directly, looking for viral DNA which signs its presence but not the viral activity. Thus, the gold standard for proving HPV-induced status remains the detection of E6 E7 RNA, signing the intra-tumoral transcriptional activity of the virus [21]. For OPSCC, p16 activity is a surrogate of HPV status due to its good sensitivity and specificity in the oropharynx [22]. However, p16 performance in other head and neck localizations has not been clearly evaluated and is not currently recommended in clinical practice [22].

The objective of this study was to investigate the incidence of HPV infection in SNSCC and its potential impact on clinical characteristics and oncologic outcomes. P16 diagnostic performances and prognostic value in sinonasal tract were also assessed.

## 2. Materials and Methods

This study was performed once approval was received from the local Research Ethics Committee, in accordance with the World Medical Association—Declaration of Helsinki—ethical principles for medical research. This was a monocentric study of prospectively collected cases in the Head and Neck Cancer Committee Database of our institution.

### 2.1. Patient Selection

All patients treated at Gustave Roussy for a SNSCC, from January 1990 to December 2020, were included. Among the samples available in formalin fixed and paraffin embedded (FFPE), we excluded the samples that previously received radiotherapy and/or chemotherapy and those that were decalcified to avoid any technical issue. Fifty-nine patients were finally included. The collected tumor samples consisted of 46 biopsies and 13 surgical specimens, all fixed in FFPE. Every histological sample was reviewed by an expert pathologist (MC) to confirm the initial diagnosis. Morphological data of keratinization and differentiation were recorded.

### 2.2. Data

We collected all clinical and follow-up data of the patients from our institutional database. For missing data, we called the patients and/or their referring physician.

### 2.3. Immunohistochemistry

For each tumor sample (*n* = 59), we performed a p16 immunohistochemistry (IHC). All 3–5 mm section samples mounted on positively charged adhesive slides (Klinipath silan printer slides) were deparaffinized and subjected to antigen retrieval using 10 mM citrate buffer (92 °C for 30 min). The slides were stained with a Ventana Benchmark autostainer (Ventana Medical Systems, Tucson, AZ, USA). A tonsil squamous cell carcinoma with high p16 expression was used as a positive control.

p16 immunohistochemistry was scored as positive if there was strong and diffuse nuclear and cytoplasmic staining present in >70% of the malignant cells. If the stain was moderate or only cytoplasmic, we considered the tumor as doubtful. All other staining patterns were scored as negative.

### 2.4. In Situ Hybridization (ISH): RNAscope

For all positive and doubtful p16 cases, an E6 E7 mRNA detection of high-risk HPV was performed. We used the RNAscope 2.0 LEICA Biosystem BOND RX research advance and the HPV-HR7 probe cocktail (HPV 16, 18, 31, 33, 35, 52, and 58) (Advanced Cell Diagnostic©) in accordance with the manufacturer’s instructions.

Ubiquitin C (a constitutively expressed endogenous gene) and the bacterial gene, dapB (diaminopimelate B), were used as positive and negative controls, respectively. A positive HPV test was defined as punctate staining observed in the cytoplasm and/or nucleus of any of the malignant cells.

### 2.5. PCR DNA

For all positive and doubtful p16, we performed an HPV DNA PCR genotyping.

Sections of FFPE biopsies were deparaffinized overnight as previously described [23]. Two molecular HPV assays are used in parallel for routine HPV detection and genotyping on FFPE biopsies, the AnyplexTM II HPV 28 (Seegene, Seoul, South Korea) (AP28) and Inno-Lipa^®^ HPV genotyping extra II (Fujirebio, Gent, Belgium) (IL) assays. The AP28 assay that distinguishes 28 HPV genotypes, by amplifying 100–200 bp fragments of the L1 gene (including 13 HR types (HPV -16, -18, -31, -33, -35, -39, -45, -51, -52, -56, -58, -59, and -68), 8 LR types (HPV -6, -11, -40, -42, -43, -44, -54, -61), and 7 genotypes reported as possibly carcinogenic (HPV -26, 53, -66, -69, 70 -73, and -82)), and the human gene β-globin was used in two different reactions for multiplex HPV molecular testing (Lillsunde Larsson et al. J Mol Diagn 2015). Undetermined or HPV negative status with AP28 was confirmed with the reference IL assay which consists of PCR amplification of a smaller 65 bp fragment of the same L1 gene using SPF10 primers sets and the ubiquitous gene human leukocyte antigen-DPB1 as internal control, followed by hybridization of specific HPV probes in a dedicated system according to the manufacturer’s instructions. The IL assay also detects and differentiates 13 HR HPV (HPV16, 18, 31, 33, 35, 39, 45, 51, 52, 56, 58, 59, 68), nine low-risk (LR) HPV (HPV6, 11, 40, 42, 43, 44, 54, 61, 81), seven genotypes reported as possibly carcinogenic (HPV26, 53, 66, 67, 70, 73, 82), and three genotypes not described as carcinogenic (HPV62, 83, 89).

### 2.6. Statistical Analysis

The distribution of the sample was analyzed with the Kolmogorov–Smirnov test. Associations between the clinico-pathological features, p16 expression, and E6/E7 mRNA expression were evaluated using Fisher-exact and Mann Whitney tests. All statistical tests were two-sided. Two groups were established. An HPV+ group including RNAscope + patients and an HPV- group including p16- or RNAscope- patients. The overall survival and recurrence free survival of both groups were determined according to the Kaplan–Meyer method. The two groups were compared with the log rank test. The agreement between two distinct detection techniques was estimated by simple percent of concordance with its 95% confidence interval. Statistical analyses were performed with the R software.

## 3. Results

### 3.1. Patients and Tumor Characteristics

Among 190 screened patients with SNSCC, 59 with available pathologic data and tumor samples were finally included in this study (Table 1). All patients had a SNSCC de novo, without any previous or synchronous inverted papilloma. Median age was 61 years (17; 90) with a sex ratio of 1.7 (37/22).

At diagnosis, SNSCC were locally advanced tumors with 84.5% (*n* = 49) of T3–T4 stages, including 53.4% T4a (*n* = 31) and 17.2% T4b (*n* = 10). Initial nodal involvement was noted for 27.6% (*n* = 16) of the patients with eight N1, six N2b, and two N2c. All cN+ patients had T3–T4 SNSCC. No primary tumor arising from the ethmoid sinus was cN+. Primary tumor locations were maxillary sinus 69% (*n* = 40), ethmoid sinus 8.6% (*n* = 6), nasal fossa 20.7% (*n* = 12), and sphenoid sinus 1.7% (*n* = 1). The 87.2% (41/47) arising from sinus cavities were locally advanced and 66% (8/12) in the nasal fossa.

The pathologic analysis showed 9 (15.5%) poorly differentiated, 31 (52.5%) mild, and 19 (32.2%) well-differentiated SNSCC, with 35 non-keratinizing and 24 keratinizing SCC.

### 3.2. Therapeutic Algorithm

Fifty-five patients were treated with curative intent. Among them, a surgical resection was performed in 43 cases followed by adjuvant radiotherapy in 40 cases (90.9%). Twenty-six patients (44.8%) received a neo-adjuvant chemotherapy (platinum-based regimen consisting in PF or TPF) with 6 (23.1%) CR, 16 (61.5%) PR, 2 (7.7%) SD, and 2 (7.7%) PD. However, two patients died during induction chemotherapy due to a disease progression and grade V toxicity. In the non-surgical group (*n* = 13), nine patients had chemo-radiotherapy, one had brachytherapy, and three patients received palliative chemotherapy.

### 3.3. Oncologic Outcomes

Considering the patients treated with curative intent, only 51 were included in the outcome analysis due to incomplete follow-up data. Median follow-up was 47 months (+/−50). Twenty (39.2%) recurrences were reported with 12 local, 3 nodal, 4 loco-regional with distant metastases, and 1 isolated distant metastasis. Ten occurred in the first six months after treatment. Thus, in this cohort, 2- and 5-year OS were 70.2% and 54.2%, respectively, with a 2- and 5-year PFS of 55.7% and 45.5%.

Among the patients who had surgery followed by radiotherapy (*n* = 40), 2- and 5-year OS were 70.7% and 54.7%, respectively, while 2- and 5-year PFS were 63.1% and 53.9%. Of note, for nine patients not eligible for surgery who received a combination of chemo and radiotherapy, we noted a 5-year OS of 33.3%.

As expected, in the whole cohort, recurrence rate was higher in patients with tumors arising from the paranasal sinus (43.5%) than from the nasal fossa (25%). The same trend was observed in the surgical group, with 35.4% (11/31) and 11.1% (1/9), respectively (*p* = 0.045). Moreover, the overall survival was significantly better for patients with SNSCC arising from the nasal fossa (*p* = 0.012). (Figure 1)

Among the 40 patients who had surgery followed by adjuvant RT, half (*n* = 20) received an induction chemotherapy. In the ICT group, four (20%) recurrences were noted with four (20%) in the non-ICT group. However, these groups were nonhomogeneous with T3–T4 tumors in the non-ICT and the ICT group representing 48.5% and 100%, respectively.

Two (25%) recurrences were observed in patients with T1–T2 SNSCC while 18 (42%) were reported for T3–T4 tumors (*p* = 0.45). Interestingly, no recurrence was noted for T1–T2 SNSCC treated by surgery followed by radiotherapy.

### 3.4. HPV-Induced SNSCC Characterization

In the whole cohort (*n* = 59), 11 (18.6%) were p16+ and 5 (8.4%) considered as doubtful (Figure 2 and Figure 3). On these 16 tumors, RNAscope^®^ was positive in nine cases with six p16+ and three doubtful tumors (Figure 4). Thus, nine SNSCC were considered as HPV-induced (HPV+) counting for 15.2% of the whole cohort (Table 2). Among these cases, PCR genotyping analysis identified five HPV16, one HPV 18, and one HPV 33.

Patients with HPV+SNSCC were younger (48.6 yrs vs. 61.1 yrs, *p* = 0.0298) with no difference in tobacco consumption (*p* = 0.477). The primary tumor site was mainly nasal fossa in this subgroup (88.9%, *p* < 10^−4^), while HPV− SNSCC were arising predominantly from the maxillary sinus (80%, *p* < 10^−4^). No significant difference was noted in TNM stage at diagnosis although early stages were more frequent in HPV+ tumors (T1–T2: 3/9 (33.3%) vs. 6/50 (12%), *p* = 0.12). Interestingly, the pathologic findings were quite similar for HPV+ and HPV− SNSCC, with no difference in keratinization (*p* = 0.72) or tumor differentiation (*p* = 0.35).

### 3.5. Prognostic Analysis of p16 Positivity and HPV Status

No prognostic value of p16 expression was observed for OS and PFS for patients treated with curative intent (*n* = 51) and in the sub-group who received surgery with adjuvant radiotherapy (*n* = 40) (Figure 5).

Among patients with SNSCC who were curatively treated, we observed better oncologic outcomes for HPV+ tumors with a significant improvement in OS (*p* = 0.022). Interestingly, in the HPV+ sub-group, no death related to the tumor was reported during the follow-up. Moreover, no patient with HPV+SNSCC was treated by exclusive chemo-radiotherapy.

Considering the patients who had surgery with adjuvant radiotherapy, the HPV+ sub-group seemed to have better OS (*p* = 0.06) and significantly better PFS (*p* = 0.049) as no oncologic event was reported for HPV+ tumors (Figure 6). In this sub-group, six patients (15%) had an HPV+SNSCC and three of them received ICT, with two CR and one PR > 50%. In the 34 patients with HPV-negative tumors, 17 received ICT with three CR, seven PR > 50%, three PR < 50%, two SD, and one PD.

## 4. Discussion

In this cohort of rare tumors, we were able to define the specific incidence and clinico-pathologic characteristics of HPV-induced SNSCC, still not clearly established in the literature. Previous studies reported the association between HPV and sinonasal cancers with an incidence ranging from 20% to 62% [3,24]. HPV detection techniques and protocols were, however, varied.

Among our patients, 15.2% of SNSCC were considered HPV-induced, based on RNAscope analysis. A Japanese study of 101 sinonasal carcinomas screened with RNAscope reported only 8.9% of HPV+ cases [25] while, in a North American cohort analysis, based on the National Cancer Database, 30% of tumors were considered as HPV positive but detection techniques were not reported [26]. These results could be explained by differences in protocols for HPV detection, types of study design, and HPV prevalence according to geographic regions [3].

To limit potential bias, we chose to include only tumor samples without any previous treatment by chemo and/or radiotherapy [27,28,29]. Moreover, for the same reason, we excluded tumor samples after decalcification as it could interfere with immunohistochemistry and molecular analysis [30].

The gold standard to assess HPV oncogenic activity is the RNA expression of the virus, either using ISH or RT-PCR [23]. Herein, HPV status was defined by RNAscope, a ISH technique. Indeed, we observed that p16 expression does not seem to be, in our series, a reliable surrogate of HPV status considering the poor predictive value with only 56% (9/16) of p16 positive or doubtful cases confirmed in ISH. Currently, doubtful p16 cases should be considered as negative, according to American College of Pathologists recommendations and ASCO guidelines for oropharyngeal carcinoma [31]. Indeed, ASCO guidelines regarding p16 testing only apply for oropharyngeal cancer while, in other head and neck cancer sites, p16 should not be routinely tested in the clinical practice. However, 60% (3/5) of our doubtful results were finally positive in RNAscope. One explanation could be that some tumor mutations, in Rb or H3K27 genes for example, may activate p16 expression independently from HPV infection [32,33]. Moreover, RNAscope screens the seven most reported HPV serotypes in SNSCC, but some rare serotypes could be involved in the p16+ cases with negative RNAscope [18,20,34]. Analyzing the literature, data on p16 prognostic value, and its link to HPV status in SNSCC had formed mixed conclusions [24,25]. Thus, the significance of p16 expression, outside OPSCC, may be used and interpretated with caution. In our cohort, p16 expression had no predictive value of oncologic outcomes, unlike in OPSCC [35,36,37].

Interestingly, the sub-group of patients with HPV-induced tumors had specific clinical characteristics: younger patients, more T1–T2 with primary tumor predominantly arising from nasal fossa. Similar findings were previously reported in the literature [27,38,39]. Pathologic characteristics regarding keratinization and differentiation appeared, though, quite comparable to HPV negative SNSCC while in OPSCC, HPV-induced tumors are usually described as non-keratinizing [40].

In our homogenous cohort of patients with SNSCC curatively treated, we confirmed the better prognosis of HPV+ tumors with a significant improvement in overall survival (*p* = 0.022), as previously noted by several authors [18,39]. Cohen et al. also observed a better recurrence-free survival (RFS) for patients with HPV+SNSCC [41]. We confirmed this trend in HPV+SNSCC, although without reaching statistical significance.

Several explanations can be hypothesized. The better oncologic outcomes could be linked to a favorable clinical presentation at diagnosis of these HPV+ tumors with earlier T stage and easier location to perform a surgical resection with clear margins (e.g., nasal fossa). Indeed, it is demonstrated that outcomes of SNSCC are well correlated to local control with surgical resection followed by adjuvant radiotherapy being the standard of care [7,42]. Tumors originating from paranasal sinuses are, usually, close to key anatomical structures such as the orbital content, the skull base, or the internal carotid artery, decreasing the feasibility and the efficacy of a surgical treatment [7]. In our cohort, tumor primary site and HPV status were highly correlated. It was, then, not feasible to analyze independently the influence of HPV infection in SNSCC prognosis. Moreover, the potential better chemo and radiosensitivity of HPV-induced tumors, as described for HPV+ OPSCC [15,40], was not assessable as all our HPV+ patients underwent surgery before radiotherapy.

Of note, in the sub-group of patients treated by surgery followed by adjuvant radiotherapy, no benefit in outcomes was observed according to HPV status. It could be, then, hypothesized that the better prognosis of HPV+SNSCC could be more correlated to a better feasibility to perform a surgical treatment than a specific, more favorable natural history.

To our knowledge, our study represents the biggest European monocentric cohort study in the literature, relying on a homogeneous therapeutic algorithm and a standardized HPV assessment by ISH, the gold standard technique [3]. Given the scarcity of SNSCC, most clinical studies are pooling cases from different series, using miscellaneous detection techniques with few validated protocols such as E6/E7 mRNA detection with ISH or RT-PCR. A significant proportion of HPV status analyses are undertaken with DNA testing alone, potentially inducing a bias with false positive cases due to contamination or bystander infection. Thus, multicentric prospective studies with established guidelines for HPV detection and therapeutic algorithms are required to confirm these findings and to evaluate the chemo-radiosensitivity of HPV+SNSCC.

Confirming higher rates of response to chemotherapy and radiotherapy for HPV-induced sinonasal tumors, as in OPSCC, could lead to de-escalation protocols. Currently, for patients with HPV+OPSCC, chemo-radiotherapy with cisplatin has been established as a standardized therapy with favorable prognosis [43]. Considering the potential aesthetic and functional consequences of sinonasal cancer resection, switching from surgery to chemo-radiotherapy (CRT) may allow a significant decrease in therapeutic morbidity [7,44]. Recently, Abdelmeguid et al. [45] reported an organ preservation rate of 40.8% for patients with locally advanced SNSCC who were treated by radiotherapy after a favorable response to induction chemotherapy (ICT). Switching to CRT instead of surgery was, in this study, decided according to the response to ICT, limiting the surgical morbidity for good responders. The specific chemo-radiosenstivity of HPV+SNSCC remains, thus, a decisive characteristic to clearly evaluate. Indeed, HPV-status could constitute a significant biomarker to tailor the therapeutic algorithm for patients with SNSCC treated with curative intent.

Our current study may have, however, potential limitations. It is, indeed, retrospective with a limited number of patients due to the rarity of the sinonasal tumors, leading to a lack of statistical power. This low number of cases hampered the feasibility of performing multivariate logistic or Cox regression analyses. Furthermore, we eliminated patients with previous chemo-radiotherapy to reduce any bias in the pathologic and molecular analyses, considering that it can induce genetic alterations [27,28,29]. Moreover, we observed no differences in clinico-pathologic characteristics or oncologic outcomes, according to HPV genotypes. Oncogenic value of specific genotypes has yet to be described in SNSCC. In our cohort, all patients had a SNSCC de novo, without any previous or synchronous inverted papilloma (IP). This clinical presentation has, however, to be investigated as SNSCC associated with IP may constitute a different sub-group with specific outcomes and HPV genotypes. Interestingly, Rooper et al. reported that high-risk HPV, even transcriptionally active, does not seem to play a major role in the transformation of IP in carcinoma [46] while Sahnane et al. observed no difference in the incidence of high-risk HPV in IP-associated SNSCC and de novo SNSCC [47]. Several studies defined EGFR mutations as another oncogenic pathway that may lead from IP to SNSCC [48,49].

## 5. Conclusions

Our findings confirmed that patients with SNSCC induced by HPV seem to have better oncologic outcomes, probably due to earlier tumor stage at diagnosis with primary location predominantly in the nasal fossa, a more suitable epicenter to perform a surgical resection with clear margins. Unlike OPSCC, p16 expression does not appear to be a good surrogate for HPV status in SNSCC. Thus, ISH remains the gold standard technique for HPV detection in sinonasal cancers. Additional studies are required to assess the specific chemo and radiosensitivity of HPV+SNSCC, leading to a potential switch in therapeutic guidelines.

## Figures and Tables

**Figure 1 cancers-14-01874-f001:**
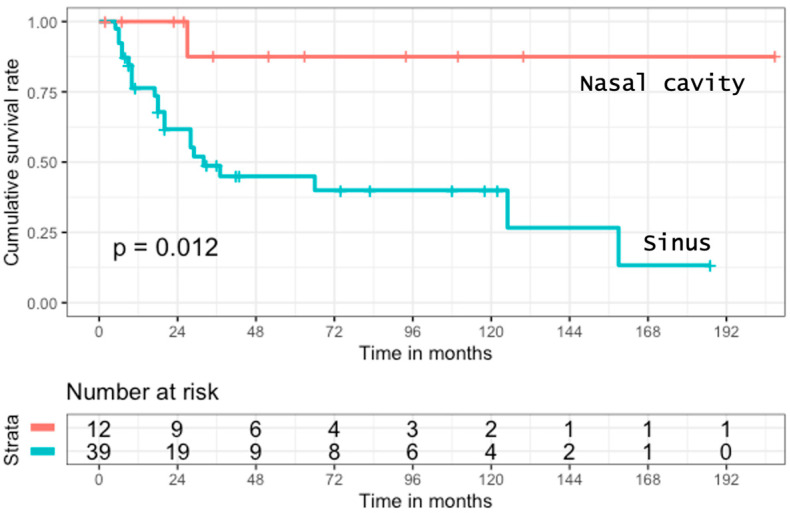
Overall survival in patients with SNSCC, curatively treated, according to primary localization.

**Figure 2 cancers-14-01874-f002:**
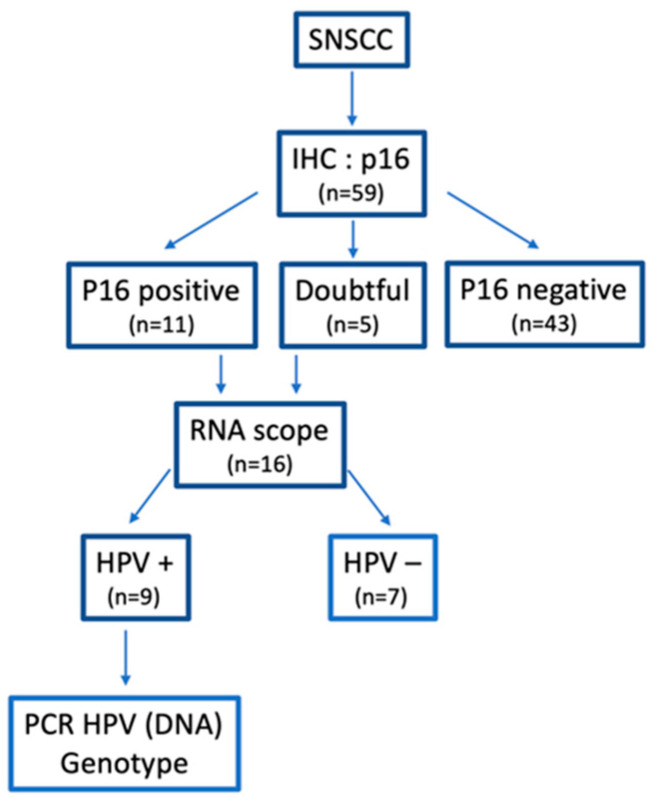
HPV status characterization.

**Figure 3 cancers-14-01874-f003:**
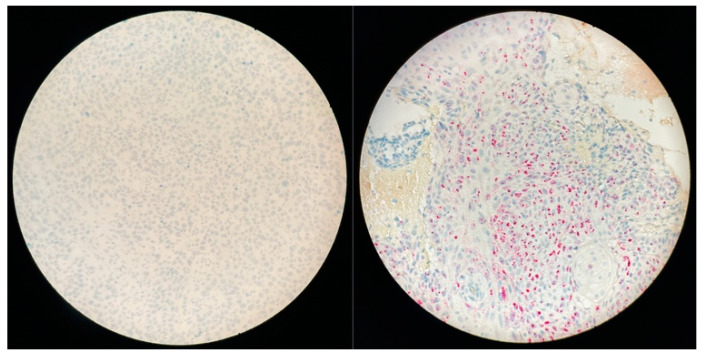
HPV RNAscope: negative (**left**); positive (**right**).

**Figure 4 cancers-14-01874-f004:**
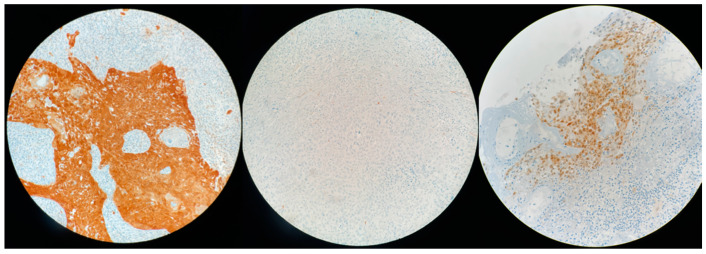
P16 expression on immunohistochemistry: positive (**left**); negative (**middle**); doubtful (**right**).

**Figure 5 cancers-14-01874-f005:**
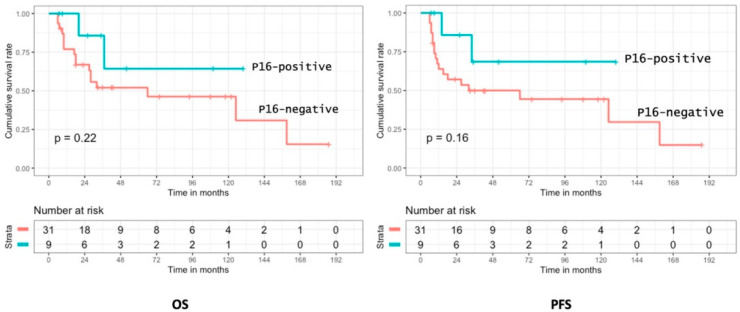
Overall survival and progression-free survival according to p16 expression in SNSCC.

**Figure 6 cancers-14-01874-f006:**
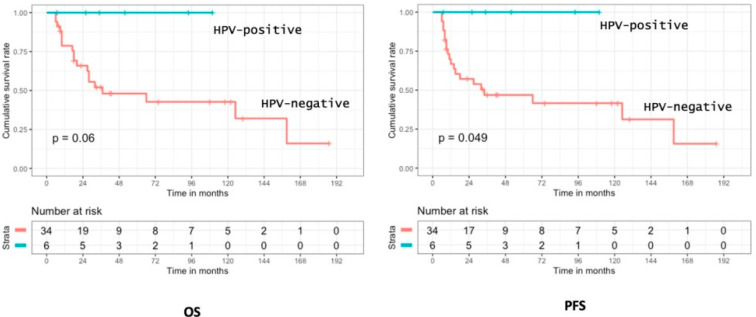
Overall survival and progression-free survival according to HPV status in SNSCC.

**Table 1 cancers-14-01874-t001:** Patient and tumor characteristics.

Characteristics	Overall Cohort
Mean age (yrs)	61 (+/−15.1)
Sex ratio (M/F)	1.7
Tobacco	
Smokers	28 (48.2%)
Mean consumption (pack years)	34.2
Primary site	
Maxillary	40 (69%)
Ethmoid	5 (8.6%)
Sphenoid	1 (1.7%)
Nasal cavity	12 (20.7%)
TNM stage	
T1–T2; T3–T4	9 (15.5%); 49 (84.5%)
N+	16 (27.6%)
M+	1 (1.7%)
Treatment algorithm in curative intent (*n* = 55)	
ICT	26 (44.8%)
Surgery	43 (74.1%)
Adjuvant RT	40 (70.7%)
CRT without surgery	9 (15.5%)
Median follow-up (months)	44.8 (+/−49.1)
Tumor recurrence	20
Local	12
Regional	3
Distant metastases	1
Multifocal (T+N+M)	4
Oncologic outcomes (patients treated curatively, *n* = 55)	
PFS at 2 and 5 years	55.7%/45.5%
OS at 2 and 5 years	70.2%/54.2%

**Table 2 cancers-14-01874-t002:** Patient and tumor characteristics according to HPV status.

Characteristics	HPV+ (*n* = 9)	HPV− (*n* = 50)	*p*
Mean age (yrs)	48.8	61.1	**0.029**
Sex ratio (M/F)	1.25	1.78	0.71
Tobacco			
Smokers	2 (22%)	24 (50%)	0.476
Primary site			
Maxillary	1 (11%)	40 (80%)	**<10^−4^**
Ethmoid	None	5 (10%)	
Sphenoid	None	1 (2%)	
Nasal cavity	8 (89%)	4 (8%)	**<10^−4^**
Pathologic findings			
Poorly; Well-Differentiated	3; 6	6/44 (12%)	0.13
Keratinizing; Non-Keratinizing	3; 6	21/29 (42%)	0.725
TNM stage			
T1–T2; T3–T4	3 (33%); 6(67%)	6 (12%); 43 (88%)	0.136
N+	2 (22%)	14 (29%)	1
M+	None	1 (2%)	

## Data Availability

Data available on request from the corresponding author.

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
