# Peer review of "Prognostic Analysis of HPV Status in Sinonasal Squamous Cell Carcinoma"

_cancers, 2022, doi:10.3390/cancers14081874_

Round 1
Reviewer 1 Report
Review
The paper addresses an important health topic for which there are still some gaps in knowledge: The HPV role in sinonasal carcinomas. However, some issues need to be addressed. I do have some comments for improvement:
Simple summary
Line 15: I would add retrospective before cohort
Line 16: Add RNA before in situ hybridization and (RNAscope) after
Line 19: I would move the sentence “No prognostic value of p16 expression was reported” after the next one indicating the prognostic value of HPV positivity
Line 20: remove “The incidence of HPV+SNSCC was 15.2%”, is redundant with sentence in line 17
Line 23: I would replace “is not” not seems to be
Abstract
For common sentences, same comments than for simple summary
Introduction
Line 71: I would add that p16 testing in HNC out of oropharynx is currently not recommended in the clinical practice
Materials and methods
Line 111: replace et by and
Results
Figure 2: You may include the numbers in each category of the flow chart
Figure 3 and figure 4: Better to change prognostic analysis by OS and DFS by p16 expression in SNSCC
Discussion
Line 260: It should be more clearly indicated that the gold standard for HPV oncogenic activity is the RNA expression of the virus, either using ISH or RT-PCR, not the ISH technique itself
Line 265: You could also discuss that the recommendations of ASCO guidelines regarding p16 testing only apply for oropharyngeal cancer, in other head and neck cancer sites p16 conditions are not well established, only that should not be routinely tested in the clinical practice
Line 273: The lack of prognostic significance of p16 can be also due to the low number of cases, that must be stated somewhere
Line 328: In the limitations sections, it should also be stated that the low number of cases hampered the feasibility of performing multivariate logistic or Cox regression analyses (if this is the case, as I presume) to assess the independent prognostic role of HPV of p16 in these series
Conclusions
As the results are based on descriptive analyses and no multivariate analyses adjusting for potential confounders were performed I would step down the significance of the results and be a bit more cautious in the conclusions
Reviewer 2 Report
The authors of this study sought to investigate the role of HPV in sinonasal squamous cell carcinoma as well as the potential applicability of p16 as a marker. Overall this study is very well done, well written, and well presented. The approach the authors take is applauded to try and standardize techniques with RNAscope followed by secondary validation. The findings the authors present have been reported by others previously, thus this study supports the growing literature. The only minor suggestive I would have for improving this study would be to add representative images as to p16, RNAscope, and HPV ISH.
Author Response
Please see the attachement.

Round 2
Reviewer 1 Report
The authors have correctly adressed the issue I raised in my review. I have not further comments.